# Design of a Portable Orthogonal Surface Acoustic Wave Sensor System for Simultaneous Sensing and Removal of Nonspecifically Bound Proteins

**DOI:** 10.3390/s19183876

**Published:** 2019-09-08

**Authors:** Shuangming Li, Venkat R. Bhethanabotla

**Affiliations:** Department of Chemical & Biomedical Engineering, University of South Florida, Tampa, FL 33620, USA

**Keywords:** surface acoustic wave, portable sensor, quartz, non-specific binding, protein fouling, biosensing, direct digital synthesis

## Abstract

One challenge for current surface acoustic wave (SAW) biosensors is reducing nonspecific adsorption. A device propagating Rayleigh and shear horizontal surface acoustic waves in orthogonal directions fabricated in ST quartz has the capability of achieving simultaneous detection and nonspecific binding (NSB) protein removal. Current measurement methods for a SAW sensor system based on this device require large-size and expensive equipment such as a vector network analyzer (VNA), signal generator, and frequency counter, which are not suitable for portable, especially point-of-care, applications. In this work, a portable platform based on a direct digital synthesizer (DDS) is investigated for the orthogonal SAW sensor, integrating signal synthesis, gain control, phase/amplitude measurement, and data processing in a small, portable electronic system. This prototype was verified for both stability and repeatability, and the results matched very well with VNA measurements. Finally, system performance in real-time sensing and NSB removal was evaluated.

## 1. Introduction

There has been a great deal of interest in developing acoustic wave-based devices for biological sensing [1,2]. A numbers of such devices have been studied for biological testing applications [3,4]. Amongst them, surface acoustic wave (SAW) devices with shear horizontal (SH) wave propagation are leaders for liquid-phase applications, due to their highly sensitive velocity and attenuation responses to perturbations from mass loading [5,6]. The guided SH-SAW device, with a waveguide layer on the delay path, would reduce the power consumption and enhance the sensitivity of the sensor [7,8]. Owing to its high sensitivity, low cost, and ease of integration with an electrical circuit, the guided SH-SAW biosensor has huge potential in point-of-care testing when integrated with a portable measurement system [9]. Such a portable biosensor system could afford the possibility of real-time monitoring at point-of-care, if issues related to the non-specific binding (NSB) of interfering proteins and incubation times are addressed.

SAW devices using acoustic wave streaming have also been utilized to provide rich information on the specificity, affinity, and kinetics of biomolecule interactions [10]. In applications demanding low limits of detection in the pg/mL range, such as for cancer biomarkers, enhancements using nanoscale physics is necessary to quantify these biomarker concentrations. One such example is given in our recent work using gold nanoprobes to construct an immunosensor for carcinoembryonic antigen (CEA) [11]. However, such a highly sensitive SAW biosensor is susceptible to background noise caused by the nonspecific adsorption of other proteins when used on a real sample such as a drop of blood. This interference from NSB proteins to the sensor signal, and long incubation times present serious challenges to achieving SAW-based point-of-care (POC) testing systems.

In our previous work, we have shown that surface acoustic wave streaming can be utilized to decrease nonspecific adsorption. Rayleigh waves with a prominent surface normal component have the ability to selectively remove NSB proteins from the surfaces of biosensors [12,13,14]. Acoustic streaming is typically generated using Rayleigh SAWs [15]. Recently, we introduced a SAW device in ST quartz capable of propagating Rayleigh and surface skimming bulk (SSBW) waves in orthogonal directions. With a suitable wave-guide bringing the energy of the SSBWs to the surface, this device has the potential for simultaneously reducing NSB protein interference and incubation times using acoustic streaming from the Rayleigh waves, while allowing for liquid phase quantification of biomarkers from the liquid phase using the wave-guided SSBWs. These concepts have been evaluated in our recent work [16,17]. Device configurations which can utilize a wave guide to bring the SSBWs to the surface, thus further enhancing sensitivity, are possible within this orthogonal configuration, and are currently being investigated in our group.

However, there is a challenge in utilizing these or any other SAW devices in POC applications. SAW devices are usually operated in the MHz to GHz range, thus the vector network analyzer (VNA) is commonly used for signal measurement. While it can provide accurate parameter testing and monitoring, the VNA is very large and expensive [18]. In addition, for the driving signal for NSB removal, a power-controllable, variable frequency signal generator is essential, but such signal generator equipment is also very heavy and expensive [19]. An inexpensive and user-friendly system to achieve sensing and removal of NSB proteins simultaneously would be essential for POC applications. Such a system should generate sinusoidal waves capable of amplification control, phase/amplitude comparison, signal detection, and communication. Chang et al. designed a surface acoustic wave aptasensor system for human breast cancer detection. They generated oscillating signals from a SAW biosensor automatically using an oscillator circuit [20]. Phase data could be directly read by a computer connected to the NI-PXI data collection system (National Instruments, Austin, TX, USA). This system did not provide a solution to replacing the commercial data acquisition equipment. Jeutter et al. provided a better method to access the phase and amplitude output of a SAW sensor operating in a liquid environment [21]. They split the 103 MHz RF signal three ways; the first one as a phase reference, the second one as an amplitude reference, and the last one as the input to the SH-SAW sensor. The output of the sensor is split two ways to measure the relative phase/amplitude differences by comparing with the reference signals. However, their portable system has to rely on an extra RF signal generator source. In this paper, we present electronics for a portable system for our orthogonal SAW device, achieving measurement of the sensor signal, and generating a signal to remove NSB protein interference.

## 2. Orthogonal SAW Sensor

All biological sensors are prone to difficulties with binding of desired proteins. This has been recognized as one of the most challenging issues found in protein patterning of biosensors. To address this problem, chemical techniques and processes have been developed, such as self-assembled monolayers (SAMs) [22,23], blocking layers [24] and zwitterionic polymers [25,26]. Compared with these traditional methods, SAW streaming enables less sample processing and easier surface modifications. Mechanisms and experimental results of acoustic streaming removal of non-specifically bound proteins have been presented in our previous publications [12,13]. Briefly, the interaction of Rayleigh waves with the fluid medium results in a wave mode conversion to leaky SAWs (Figure 1a). These leaky SAWs propagate along the boundary between the piezoelectric solid and liquid media and excite longitudinal waves into the fluid at the Rayleigh angle θ. Direct SAW forces result in the initial NSB particle detachment, whereas hydrodynamic forces (drag and lift) prevent their reattachment. As the NSB forces caused by weak interactive forces (i.e., Van der Waals and hydrophobic) are typically much weaker than the specific binding forces between antibodies and antigens, the wave streaming is able to remove only the NSB proteins without detaching the specific antibody-antigen links with a modest energy input.

While Raleigh waves are very useful for generating acoustic streaming forces for NSB protein removal, they are unsuitable for biosensing in liquid media, for which shear-horizontal (SH) polarization is necessary to avoid damping the wave. ST quartz is capable of supporting Rayleigh and SSBWs with SH polarization in orthogonal directions, as shown in Figure 1b. With the application of a wave guide, the SSBWs are converted to Love waves, providing for a sensitive biosensing platform. In this work, this orthogonal device is used for both sensing and NSB protein removal.

The dual function SAW sensor was realized by placing two pairs of interdigital transducer (IDT) electrodes on the same ST-quartz substrate (Figure 1c). To obtain a guided wave, a gold layer is added along the delay path of the sensing direction. An O-ring cell was fabricated on the center of the chip as the sensing area utilizing SU-8 photoresist.

The orthogonal SAW devices were fabricated on 4 inch, 0.5 mm thickness, and single-side polished ST X quartz wafers. Each of the IDT electrodes for the sensing and removal directions consisted of 60 finger pairs with an electrode width of 10 µm and wavelength of 40 µm. The delay path for the sensing direction is of 8 mm length and 2 mm width, which is coated with a 100 nm gold film as a waveguide layer. No reflecting gratings were applied in this work and it is a specific set-up design for biotesting application with an O-ring cell. These patterns were fabricated by the following steps: First, NR9 1500PY (Futurrex) negative photoresist was applied by spin coating on the wafer after solvent cleaning. After the pre-bake, the coated wafer was exposed to broadband UV light using an EVG-make mask aligner, followed by a hard bake. The pattern was developed in RD6 (Futurrex) developer for 12 s, followed by rinsing with DI water and drying with nitrogen gas. E-beam evaporation was used to deposit 20 nm/100 nm Ti/Au adhesion and metal layers. The deposition rate was set to 0.5 nm/s for Ti deposition and 1 nm/s for Au to obtain strong adhesion between the substrate and metal layer. An acetone bath was used to lift-off the metal and the remaining metal pieces were removed with solvent cleaning, with ultrasonication used as needed to achieve complete cleaning. After the metal patterns were fabricated, the wafer was spin-coated with SU-8 50 negative photoresist for a similar lithography process, to obtain an O-ring cell of 2 mm diameter and approximately 100 micron height. The patterned wafers were diced into 25 × 25 mm individual chips.

## 3. Portable System Design

There are two commonly utilized methods for measuring the SAW device response [27]. One is frequency detection, as shown in Figure 2a. This basic testing system consists of an RF amplifier for a feedback loop and a frequency counter. The electronic circuit should satisfy the Barkhausen stability criterion: (1) The loop gain is equal to unity in absolute magnitude, that is, |*β**A***| = 1; and (2) the phase shift around the loop is zero or an integer multiple of 2π, that is, entire phase shift = 2πn (n = 0, ±1, ±2, ····), upon which, the entire circuit would oscillate as a resonator around the center frequency of the SAW device [28]. Though this method only requires very simple circuit design, the system is not stable and is easily affected by the environment, which usually causes frequency hopping. As there could be many frequency points that satisfy the oscillation starting conditions, once the SAW is affected by a big perturbation, the circuit will self-oscillate at a different frequency. In addition, this system also requires a high-speed frequency counter that is not easily miniaturized for portable application. The other method is phase detection using a phase detector, as shown in Figure 2b, comparing the input/output difference of the SAW device [21]. The system gets an RF source with a certain frequency and splits the signal two ways, one passes through the SAW device and the output signal is compared with the other signal. The phase difference as a voltage output can be easily measured by a voltmeter. Since it is not based on feedback, this detection system is not susceptible to frequency hopping and would be very stable. However, to achieve simultaneous sensing and NSB removal, there will be a problem of supplying a RF signal source for both the sensing and removal inputs. Direct digital synthesis (DDS) [29] would be a good method of producing an analog waveform by utilizing a time-varying signal in a digital form and converting using a digital-to-analog converter (DAC). It can offer fast switching of frequency, a wide output bandwidth, and very high-frequency resolution. Thus, we designed a DDS based system with time division multiplexing for this dual function sensor system.

### 3.1. Overall System Design

The electronic circuit system consists of three boards and is assembled as a portable instrument, as shown in Figure 3. The bottom layer is based on the Arduino development board with Intel^®^ Curie™ Module. This layer provides the microcontroller for the main program operating the system. It provides serial communication with the local computer and the control with other IC chips. The middle layer, which is the main circuit board, is designed for the power supply, DDS signal generation, signal processing, and data acquisition/converting, etc. The top layer is a specially designed PCB board as a chip holder for the SAW device with connecting clips and buffer circuit.

To realize a real portable system, some essential peripheral designs are added to this system. The power supply of this system can range from 7–12 V which is easily obtained from a commercial power adapter. Communication between the portable prototype and compute is via USB cable. The data communication, storage, processing and displaying are conducted with the self-developed SAW sensor data monitoring software. The SAW device is loaded on a specially designed board with eight clips touching with each electro pad of the SAW sensor. This board is connected with the main circuit board via SubMiniature version A (SMA) connectors of 50 Ω impedance. In addition, these electronics are housed in a 3D printed shell package. The dimensions of the portable device are 110 × 110 × 80 mm.

### 3.2. Main Circuit Design

Figure 4 shows the designed electronic system to measure phase, transmission loss, and frequency for the SAW sensor device. The RF signal is generated from the 32-bit DDS chip (AD9858, ADI, Norwood, MA, USA) and the working frequency is calculated and set by the microcontroller. This synthesizer is able to offer high resolution of 0.233 Hz and a wide bandwidth sine wave up to 400 MHz with special designed temperature compensated crystal oscillator (TCXO). The TCXO provides a stable reference with frequency stability of 0.5 ppm at 1 GHz, which is eligible for biotesting at room temperature. The sine wave is amplified by a digital controlled variable gain amplifier (DVGA), which can provide a +19 dB gain, with final output power about +16.4 dBm. The internal-integrated digital controlled attenuator of this amplifier provides a power attenuation coefficient ranging from 0 dB to −31.5 dB (the final gain is from −12.5 dB to +19 dB). After amplification and low-pass filtering, the RF signal is delivered into two channels controlled by the RF switcher, according to the purpose of use. One channel is separated via the 2-way 0° power splitter, and utilized to load the two sides of the removal IDTs. The other channel is separated into two paths via the power splitter as well, with one signal sent to the SAW sensing device as the input signal for sensing, and the other signal as one input source of the gain/phase detector as a reference signal. The output signal of the SAW device, as another input source, is compared with the reference signal via the gain/phase detector. The output voltage of the gain/phase detector, as a function of the two input signals’ amplitude/phase differences, represents the insertion loss and phase shift of the SAW device. The voltage value is obtained by the microcontroller via the 12-bit analog-to-digital converter (ADC), which offers a phase angle shift resolution of 0.044° and insertion loss shift of 0.0147 dB, respectively. Eventually, after the processing and calculations, the data are received by the local computer device.

## 4. Experimental Sections

### 4.1. Reagents and Apparatus

All chemicals were of analytical grade and were used as received. 11-Mercaptoundecanoic-acid (MUA), *N*-(3-Dimethylaminopropyl)-*N*′-ethylcarbodiimide (EDC), N-Hydroxysuccinimide (NHS), Rabbit IgG, bovine serum albumin (BSA) were purchased from Sigma-Aldrich. Mouse anti-rabbit IgG was purchased from Santa Cruz Biotechnology.

The instruments utilized in the experiments were Agilent 8753ES S-parameter network analyzer and Agilent 54616B oscilloscope.

### 4.2. Sensor Surface Preparation

The surface of the SAW device was solvent cleaned and treated by O_2_ plasma. Surface modification based on a SAM and carbodiimide chemistry was employed in the study. Gold-coated disks were modified by thiols. The gold waveguide layer was incubated for 2 h with 10 mM MUA in pure ethyl alcohol, then rinsed with pure ethyl alcohol and dried by N_2_ gas. The activation solution (200 mM EDC and 50 mM NHS in deionized water) was then added to activate the carboxyl group for 10 min at room temperature. Then, protein A was assembled on the surface of the SAW device and stored overnight, followed by 200 μg/mL rabbit IgG with the same procedure. Then, 1% BSA solution in PBS was used to block the non-sensing surface (treatment time at least two hours), then rinsed with PBS solution and dried using N_2_ gas. This modified SAW biosensor was stored at 4 °C.

### 4.3. Biodetection Strategy

The modified SAW chip was tested for IgG sensing as followed (Figure 5). First, the system was set working in sensing mode, and removal channel frequency was set on its center frequency. The phase shift was monitored. 10 μL PBS solution was added in the O ring cell to obtain the baseline. After drawing up the PBS solution, 10 μL mouse anti-rabbit IgG of 10 μg/mL was added. Once the phase shift became stable, a 10 μL solution of 10 mg/mL BSA in PBS was added as the nonspecific proteins. After the phase shift became stable again, the removal RF power was switched on to separate the nonspecifically bounded proteins from the surface, then removal power was turned off.

## 5. Results and Discussion

### 5.1. Circuit Performance Test

The DDS system output performance was evaluated using Agilent 54616B oscilloscope, to measure the output sine wave signal frequency and voltage. The output port of the DDS system was connected to the oscilloscope via a 50 Ω cable. Figure 6a,b show the DSS output signals at 100 MHZ and 400 MHz after amplified, and we see that the DDS system is able to synthesize very stable sine waves. Even though our SAW devices working frequency is usually not higher than 150 MHz, we still tested for a wider range, from 20 MHz to 200 MHz, at various power levels by adjusting the power attenuation coefficient of the digital gain amplifier. It shows that the output voltage has a slight drop with increasing frequency. The output voltage also decreases when increasing the power attenuation coefficient of digital gain amplifier. The decreasing trend perfectly matches with the setting power damping ratio (dB), according to the theoretical calculation rule between voltage gain and power gain.

The circuit without the influence of SAW device was tested for 10 min (sampling rate of 0.5 Hz) to assess the system stability, as shown in Figure 7. The SAW sensing channel was electronically shorted using the 50 Ω cable. The DDS output frequency was tested at 50 MHz, 100 MHz, and 200 MHz, respectively. The ADC results only vary within 1–2 least significant bit (LSB), corresponding to lower than 0.088 degree/0.0293 dB in phase angle/amplitude, which indicates that the designed circuit has excellent stability.

The prototype with a SAW device loaded was tested using frequency scanning mode and compared with the VNA results. The prototype was set under the network analyzer mode first, measuring the amplitude-frequency dependence. Figure 8 shows the three times measurement results of the orthogonal SAW device and the S21 parameters of the same device measured by the VNA. The three measurements almost overlap one another (Figure 8a,c), showing the prototype has high reliability and repeatability. The sensing channel has an average center frequency of about 118.22 MHz, and the removal channel of about 77.66 MHz, which perfectly match with the VNA results (Figure 8b,d). The relative difference in amplitude might be caused by the systematic bias of amplifier and circuit wire, which is not important for sensing. The distorted frequency responses of VNA and prototype results could be due to the SU-8 O-ring cell on the top of the device surface. The phase vs. frequency comparison is excellent and is presented in Appendix A.

### 5.2. Dynamic Response

The shift of the phase angle of the SAW device sensing path as a function of time is sensitive to the loading on the surface. The prototype works in the sensing mode on the center frequency, which was measured under the network analyzer mode before. The dynamic response of the system was tested by dropping water on the SAW sensor (Figure 9). The baseline without liquid was measured first for 2 min. Once the system became stable, 10 µL of water was loaded in the O ring cell of SAW sensor (30 s), then, the water drop was drawn up using a pipette, after the data became stable again (60 s). This process was repeated seven times to check the system repeatability. As the result shows, the system can rapidly respond to the loading, with about 2.5° phase angle shift, and recovers to the initial level after drawing up the water and the surface is completely dry, indicating excellent system repeatability. The amplitude response was tested as well, and the device only has a very slight shift (less than 0.3 dB) after liquid loading, because the SH-SAW has very little energy consumption in liquid phase operation [6].

Our previous experimental and numerical studies indicate that Rayleigh acoustic wave streaming can cause viscous heating of a liquid droplet on the surface of the SAW device [30]. This heating might influence the sensing and can be a drawback during NSB removal. However, this heating effect is rather weak with ST-quartz compared to other piezoelectric substrates such as 128° YX LiNbO_3_. The ST-quartz device is only heated up by 0.7 °C [31]. To evaluate its influence, the orthogonal chip was tested with sensing and removal signal working simultaneously. First, the orthogonal SAW chip was set working in sensing mode, and removal channel frequency was set to its center frequency. Then, the phase shift was monitored during the removal power was on and off. As shown in Figure 10, the phase angle became stable after the water was loaded (20 s). The removal power was switched on at the moment of 120 s and then turned off after 60 s. After the moment of 240 s, this process was repeated. It is observed that neither removal power on nor off would cause a significantly observable effect on the results of the sensing path.

### 5.3. IgG and BSA Test Response

Preliminary biotesting was performed for general IgG detection. 10 μL mouse anti-rabbit IgG of 10 μg/mL was added on the modified SAW sensor and the phase shift was recorded as shown in Figure 11. The IgG molecules assembling on the surface could result in phase decrease. After subtracting the PBS solution baseline change, the phase was significantly changed after the sample was added. It finally reached around 3° within 5 min, indicative of gradual binding on the sensor surface.

Additionally, the NSB removal feature of this orthogonal SAW sensor was evaluated by loading with BSA solution. 1 mg/mL BSA in PBS solution was utilized as NSB protein in this test, which was added on the modified SAW sensor. As shown in Figure 12a, a slight phase shift about 0.4° was detected, owing to the adsorption on the sensing surface within 120 s. This adsorption force was not strong enough to hold the BSA particles once the removal SAW streaming force was turned on. As indicated in Figure 12b, after the removal power was turned on (15 s), the non-specifically bounded BSA molecules left the sensor surface resulting in a phase increase. The phase recovered to the initial level corresponding to the mass adsorbing after the removal power was switched off (75 s). A perturbation was observed when the removal streaming was switched on/off, which could be a result from the thick BSA solution exhibiting non-Newtonian flow behavior.

## 6. Conclusions

A portable system for the orthogonal SAW sensor is developed in this work. This prototype is able to achieve biodetection and NSB removal in real-time measurements. The system has shown excellent performance in both phase and insertion loss testing. Such a portable system has a great advantage in integrating vector network analyzer and signal generator in a small size prototype, which implements a significant small, light-weight, low-cost, low-power detection instrument. Furthermore, this system has proven to perform well for liquid biomolecule monitoring and has a great potential for point of care testing, with the indicated next step of the development of a biosensor for POC operation directly from body fluids such as blood and urine.

## Figures and Tables

**Figure 1 sensors-19-03876-f001:**
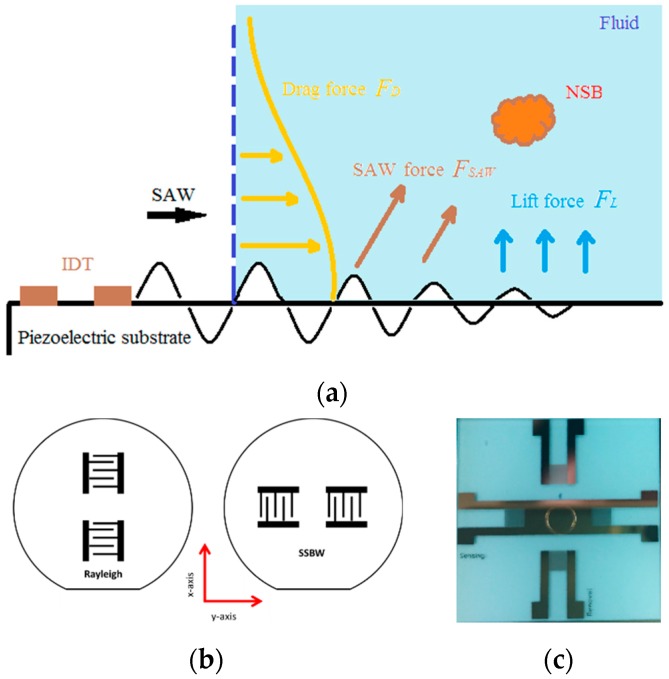
(**a**) Schematic diagram of NSB protein removal using acoustic streaming forces generated by the Rayleigh wave device; (**b**) different wave modes in the orthogonal device in ST quartz; (**c**) photograph of an orthogonal surface acoustic wave (SAW) chip.

**Figure 2 sensors-19-03876-f002:**
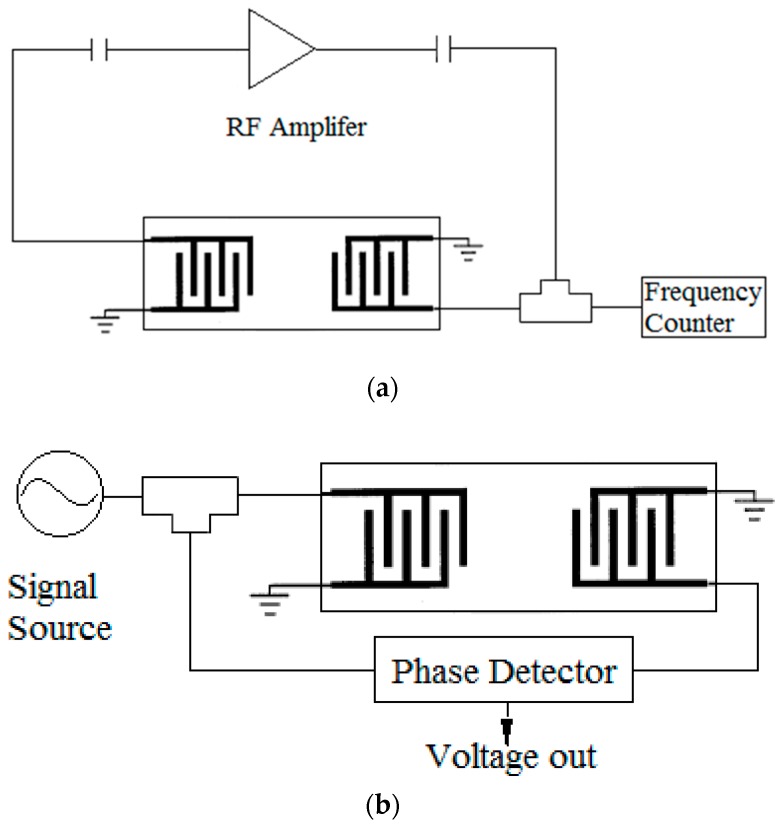
Schematic of two commonly methods for SAW device measurement: (**a**) Frequency detection of the SAW oscillator; (**b**) phase detection of SAW with a signal generator and a phase detector [21,27].

**Figure 3 sensors-19-03876-f003:**
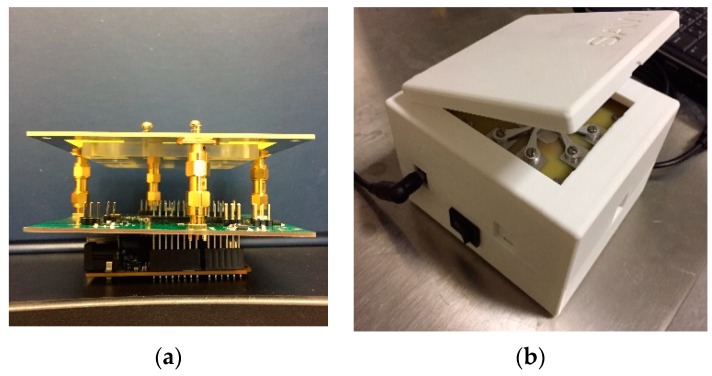
(**a**) Portable system circuit boards; (**b**) 3D printed packaging.

**Figure 4 sensors-19-03876-f004:**
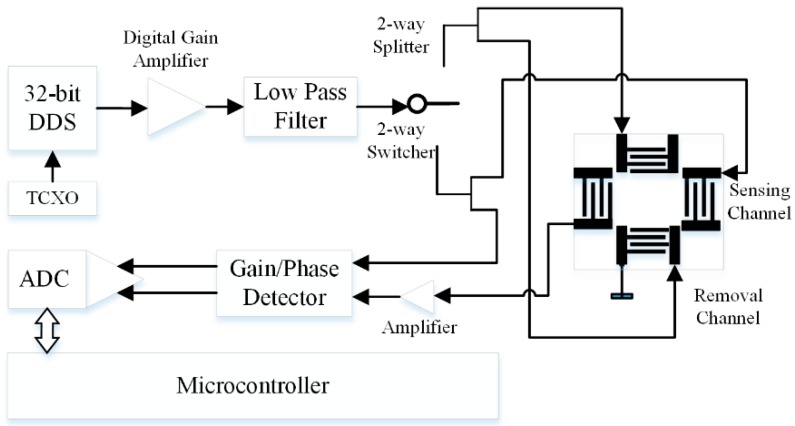
Schematic of direct digital synthesizer (DDS) based orthogonal SAW system electronics.

**Figure 5 sensors-19-03876-f005:**
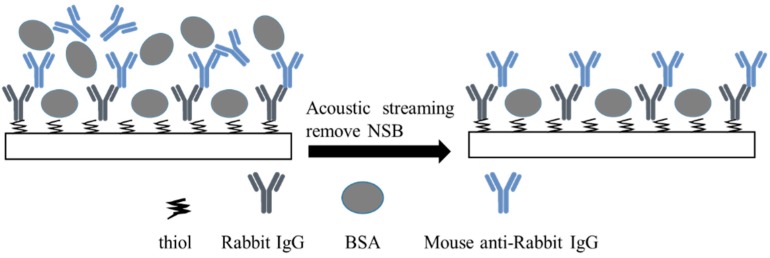
Schematic of SAW sensing and nonspecific binding (NSB) removal strategy.

**Figure 6 sensors-19-03876-f006:**
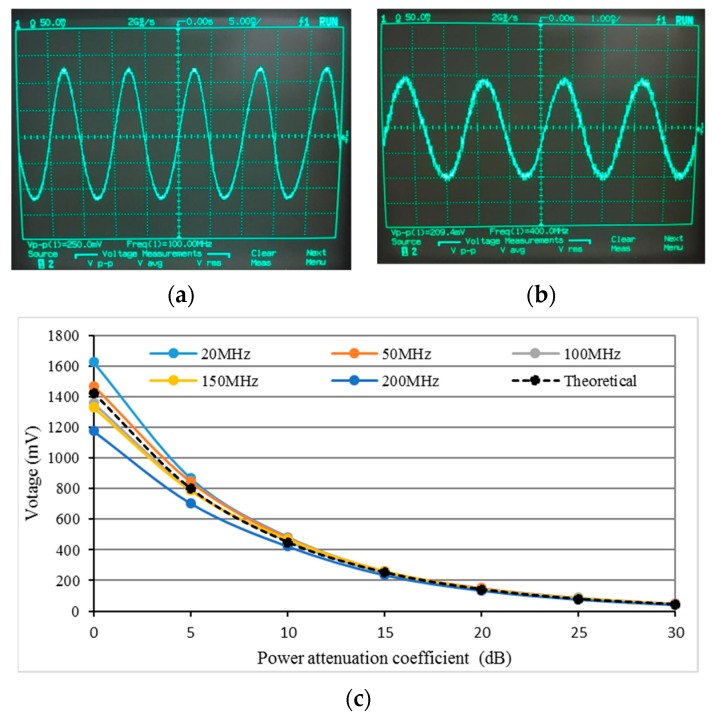
(**a**,**b**) Oscilloscope measurement results at 100 MHz and 400 MHz, with digital gain amplifier’s power attenuation coefficient of 15 dB; (**c**) Output voltage in different frequencies vs. power attenuation coefficient of digital gain amplifier.

**Figure 7 sensors-19-03876-f007:**
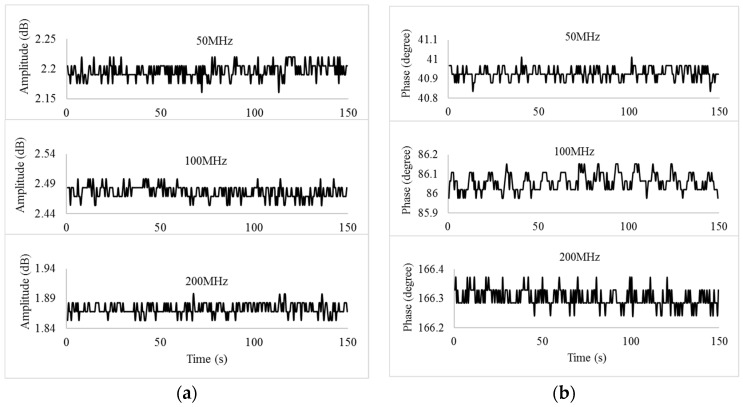
Circuit stability test at different frequencies: (**a**) amplitude; (**b**) phase angle.

**Figure 8 sensors-19-03876-f008:**
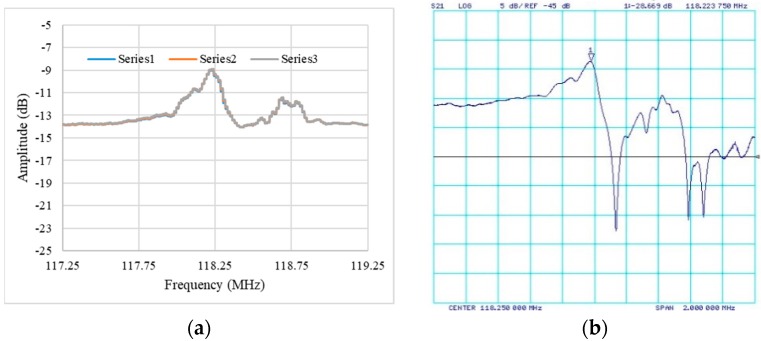
Comparison of amplitude vs. frequency results with vector network analyzer (VNA) measurement: (**a**,**b**) Sensing path and (**c**,**d**) removal path; left for portable prototype and right for VNA. The series 1–3 are triplicate measurement results of the orthogonal SAW device.

**Figure 9 sensors-19-03876-f009:**
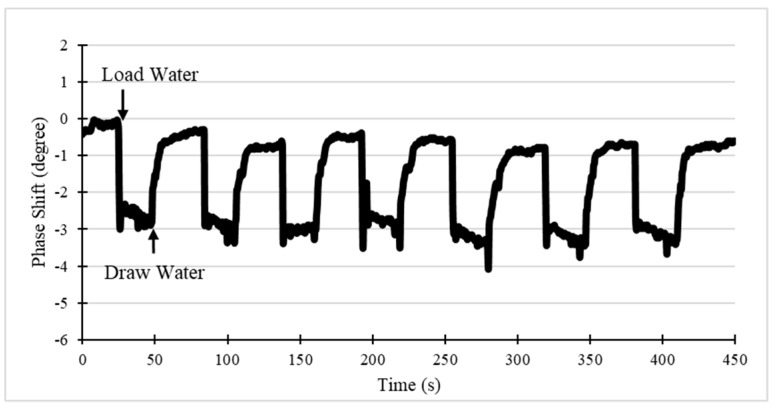
Phase response of the SAW device to water drop loading and drawing.

**Figure 10 sensors-19-03876-f010:**
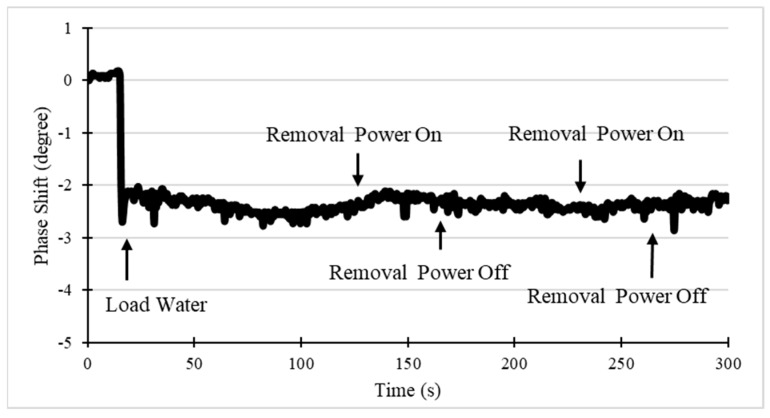
Phase response of the SAW device to the removal power switching on/off.

**Figure 11 sensors-19-03876-f011:**
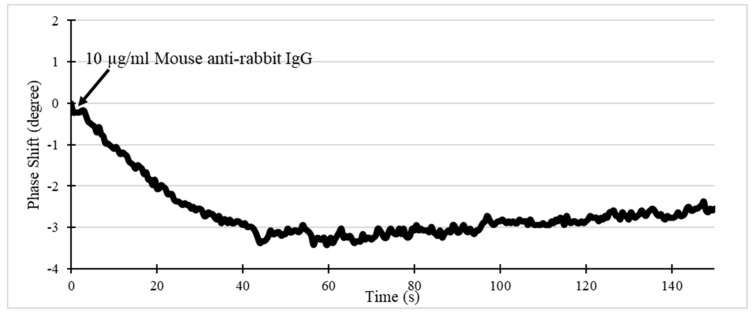
Phase response of the SAW device to mouse anti-rabbit IgG binding. Note that the relative phase shift is compared to the PBS solution baseline.

**Figure 12 sensors-19-03876-f012:**
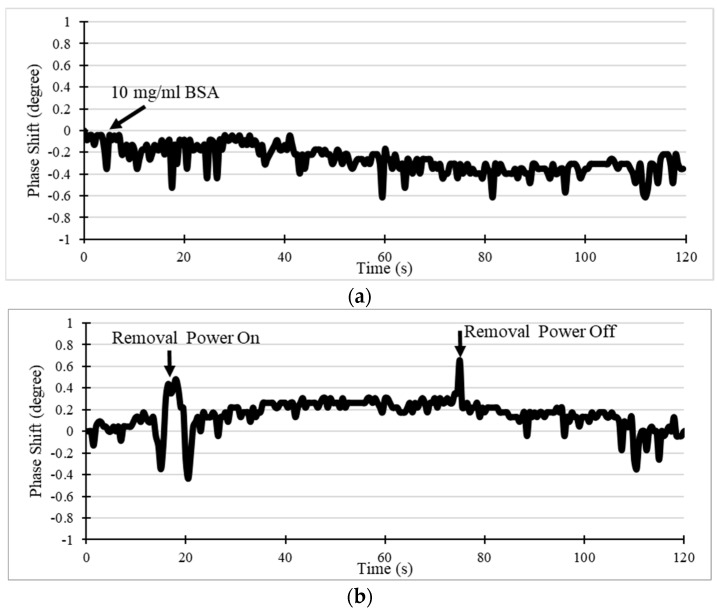
Phase response of the SAW device to (**a**) bovine serum albumin (BSA) non-specifically binding on the surface and (**b**) RF power on/off for NSB removal.

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
