# Peer review of "Design of a Portable Orthogonal Surface Acoustic Wave Sensor System for Simultaneous Sensing and Removal of Nonspecifically Bound Proteins"

_sensors, 2019, doi:10.3390/s19183876_

Round 1
Reviewer 1 Report
This article describes the development of an orthogonal Surface Acoustic Wave sensor for the sensing and removal of non specifically bound proteins. One of the significant points of the work is the possibility to remove non-specific bound proteins from the surface of a SAW immunosensor. Some questions and observations should be taken into consideration.
-What is the influence of the volume of solution on the signal variation? Could the column of liquid on the sensors significantly change the signal or is the change mostly on a surface level?
-Section 5.2 mention heating as a result of Rayleigh acoustic wave streaming. It should be mentioned that the ST X quartz device is only heated up by 0.7 °C as the reference shows to better illustrate that the sensor is not as harmful to immobilized biomolecules as sensor with other substrates.
-For the measurements carried of in figures 9, 10 and 11 the values of the phase variation should be discussed in the text in more detail, specifying values and their variations. The differences in phase variation could also be expressed as percentages. It should be clear what are the significant figures for a SAW signal.
-Where experiments carried out at other mouse anti-rabbit IgG concentrations? The main focus of the article is a proof-of-concept for a modified SAW sensor and mouse anti-rabbit IgG is an example but it would still be useful to have a linear range and calculate the other performance parameters to better illustrate the sensitivity of this type of sensor.
-Some information about the reusability and stability of the sensor are necessary. How was the sesnor regenerated after the antibody binding and how many times can one sensor be used for the detection? What is the signal variation after a number of measurement of antibody and several removal procedures?
-Figures 9, 10 and 11 should show the ticks corresponding to the values on each axis. In their current form the graphs are confusing and make it difficult to estimate values of points on the graph.
Reviewer 2 Report
The paper reports a portable system based on acoustic wave biosensors with removal of nonspecifically bound proteins. The core element of the portable system is a direct digital synthesizer combined with a gain and phase detector. The issue of the manuscript is of great interest for future biosensors based on SAW devices, therefore the manuscript could be valuable to publish in Sensors, however a major is required in order to improve its impact.
1.- (line 11) “with an electrode width of 10 µm and wavelength of 4 µm. Is the wavelength 40 µm?
2.- (lines 130-131) “the feedback loop satisfies the following conditions: 1) Lop gain>1 and 2) total loop phase = 2πn (n= 0, ±1, ±2, …)”. Conditions or criterion??? Are the mentioned “conditions” real Barkhausen criterion? Please write correctly this part of the manuscript.
3.- Please add in the manuscript frequency response of the sensor and removal SAWs with the same spam to compare de signals.
4.- It is understanding that in an experiment the synthesizer is first tuned for sensing then for removal, and finally for sensing again. In Figure 7 the real time circuit stability was studied, however a study to determinate the stability of the system in the tuned process is required.
5.- The author compare only amplitude with VNA, however, is it possible for sensor to characterize the frequency response of the phase with VNA and portable system? In this case the slope phase must be the same. Once the slope of the phase is calculated it is very interesting compare the resolution of the designed system with the mentioned in the Fig. 2 a) “frequency detection of SAW oscillator” it can be found in the state of the art.
6.- In the text (lines 281-284) the times seem not to be in coincidence with the figure 10.
7.- The explanation of the figure 12 is not clear, because responses can be confused with the tendency of the signals. However, this result is not significant for the paper, therefore in the absence of better example it can be kept.
Round 2
Reviewer 2 Report
The authors responses were ok, whoever some responses (response 3 and 5) were no introduced in the text.
Please verify it was added correctly in the manuscript.
